# Cold-Induced Nuclear Import of CBF4 Regulates Freezing Tolerance

**DOI:** 10.3390/ijms231911417

**Published:** 2022-09-27

**Authors:** Wenjing Shi, Michael Riemann, Sophie-Marie Rieger, Peter Nick

**Affiliations:** Molecular Cell Biology, Botanical Institute, Karlsruhe Institute of Technology, Fritz-Haber-Weg 4, 76131 Karlsruhe, Germany

**Keywords:** C-repeat binding factor 4 (CBF4), cold stress, grapevine, *Vitis vinifera*, jasmonate, nuclear import, proteasome

## Abstract

C-repeat binding factors (CBFs) are crucial transcriptional activators in plant responses to low temperature. CBF4 differs in its slower, but more persistent regulation and its role in cold acclimation. Cold acclimation has accentuated relevance for tolerance to late spring frosts as they have become progressively more common, as a consequence of blurred seasonality in the context of global climate change. In the current study, we explore the functions of CBF4 from grapevine, VvCBF4. Overexpression of VvCBF4 fused to GFP in tobacco BY-2 cells confers cold tolerance. Furthermore, this protein shuttles from the cytoplasm to the nucleus in response to cold stress, associated with an accumulation of transcripts for other CBFs and the cold responsive gene, ERD10d. This response differs for chilling as compared to freezing and is regulated differently by upstream signalling involving oxidative burst, proteasome activity and jasmonate synthesis. The difference between chilling and freezing is also seen in the regulation of the *CBF4* transcript in leaves from different grapevines differing in their cold tolerance. Therefore, we propose the quality of cold stress is transduced by different upstream signals regulating nuclear import and, thus, the transcriptional activation of grapevine *CBF4*.

## 1. Introduction

Global warming, as one of the most fatal aspects of climate change, is influencing all the life stages of land plants. As revealed by a study on 500 plant taxa in Massachusetts (USA) and 400 British plant species, flowering has advanced since 1736 by an average of 4–7 days with each °C increase, leading to precocious bud breaks driven by elevated temperature [1]. Additionally, extreme weather phenomena, such as early autumn frost, late frost in spring, as well as cold and hot waves during the early summer occur more often, which leads to an aggravating negative impact on plants. For viticulture, the problem is that the freezing tolerance of grapevine drops rapidly after bud break [2]. The progressively blurred onset of spring, with unusually warm weather in March and late freeze episodes in April has, therefore, especially devastating consequences for viticulture. Thus, understanding the molecular mechanisms underlying cold stress and cold adaptation has turned into a very relevant research topic.

The quality of cold stress is strongly dependent on the temperature and, traditionally, two versions of cold stress are discriminated (for classical reviews see [3,4]): chilling stress is defined as chilling temperatures above the freezing point (0–20 °C) and affects the productivity and quality of plants inhabiting tropical or subtropical climes [5]. Unlike chilling, the injury caused by freezing stress (<0 °C) is irreversible and can be lethal due to the formation of intracellular ice crystals. Whether a given plant is tolerant to chilling or freezing is strongly dependent on its genotype. In the case of grapevine, *V. amurensis* can survive at −40 °C, while severe cane damage of *Vitis vinifera* already starts at +2.5 °C for Riesling and at −5 °C for Pinot Noir, according to the USDA extension service (2021). The ability to respond adequately to different temperatures obviously requires efficient and specific signalling.

The primary input for sensing and transduction is a membrane rigidification, i.e., a physical signal requiring transformation into a chemical signal (so-called susception) [6]. The ability to discern different levels of cold stress indicates that the sensor must be able to measure time. A swift, but transient elimination of cortical microtubules has been identified as a necessary element for cold acclimation in winter wheat [7] and grapevine cells [8]. In both systems, a transient treatment with pronamide, a mild microtubule disruptor, can mimic the effect of chilling with respect to subsequent cold acclimation. The transduction itself involves Ca^2+^ influx triggered by a conformational change of the membrane-located influx channel. This seems to be facilitated (at least in rice, which is a species very sensitive to chilling) by the transmembrane protein COLD1, a regulator of the plant trimeric G-protein RGA [9]. In addition, oxidative burst via a membrane-located NADPH oxidase is a necessary element of early transduction [10]. Likewise, kinase signalling, for instance, through COLD-RESPONSIVE PROTEIN KINASE 1 [11], MAPK cascades [12], or Open Stomata 1 [13] conveys signals to the nucleus, converging on the transcription factor inducer of CBF expression 1 (ICE1). Most of this kinase signalling is acting as negative regulator and, thus, seems to be involved in the modulation of signalling rather than signalling itself (for a detailed discussion see [6]). From timing and effect, Open Stomata 1 might be the most relevant candidate for signal transduction per se.

The next event in signalling is the accumulation of ICE1, a bHLH transcription factor, which, in the absence of cold stress, is continuously degraded in the proteasome. This proteolytic decay is blocked under cold stress, such that the ICE1 protein accumulates (for review see [14]). The turnover of ICE1 is positively and negatively regulated by many factors involved in cold-induced posttranslational modifications, such as sumoylation by SIZ1 and SIZ2, ubiquitination by HOS1, and phosphorylation by OST1/SnRK2.1, or the protein kinases MPK3/6 [15]. This master switch then activates cold-box factors (CBFs) and transcriptional activators which are rapidly and highly induced by cold stress, which then turn on the expression of COR genes by binding to their C-repeat/dehydration-responsive (CRT/DRE) cis-elements in their promoters [16]. In *Arabidopsis*, approximately 170 COR genes are CBF-dependent, involved in different adaptive responses, such as the re-fluidisation of membranes, osmotic adjustment by compatible solutes, or protection of membranes and proteins by cryoprotective proteins and soluble sugars [17].

Thus, the CBFs play a key role as the second tier of transcriptional activation. The *Arabidopsis* homologues CBF1, CBF2, and CBF3, also known as DREB1b, DREB1c, and DREB1a, are rapidly induced in response to cold stress [18] and can, upon overexpression, effectively activate downstream genes, thus enhancing their cold response [19]. Conversely, the *A. thaliana* null *cbf1 cbf2 cbf3* (*cbfs*) triple mutant showed an impaired chilling response and is extremely sensitive to freezing, even after a cold acclimation treatment [20]. The second tier of signalling allows not only for signal amplification, but also for signal diversification. In fact, loss-of-function mutants showed that CBF1 and CBF3 are required for freezing tolerance as result of cold acclimation, while CBF2 was dispensable in this respect and, thus, might convey alternative functions [21]. Furthermore, *AtCBF2* is expressed in a pattern that is different from *AtCBF1* and *AtCBF3* [21]. In many (but not in all) plant species, signal diversification is also manifested as a fourth member of the CBF family. This CBF4 differs in its kinetic behavior. Its induction is slower than it is for the other CBFs, but persists longer, for instance, in leaves and buds of grapevine [22]. The function of CBF4 is, therefore, linked to cold acclimation. Indeed, constitutive overexpression of VvCBF4 produced grapevines with superior freezing tolerance [23].

In our previous work [8], we found a correlation between the expression of grapevine CBF4 and the acclimation to cold stress. To test whether this correlation derives from a causal relationship, we generated transgenic lines in the cellular tobacco model BY-2 that overexpressed CBF4 from the Pinot Noir grapevine variety in fusion with GFP to test whether these cells show constitutive cold tolerance. We observe that CBF4-GFP resides in the cytoplasm, but shuttles to the karyoplasm in response to freezing stress, accompanied by improved cellular resilience. We show that chilling and freezing stress differ with respect to the induction of transcripts for cold-box factors and cold responsive (COR) genes, including the foreign CBF4 and its native tobacco homologue *Avr9/Cf9*. This is reflected by differences in early signalling, monitored by differential transcript responses to specific inhibitors. In particular, inhibition of nuclear import, inhibition of protein synthesis de novo, and inhibition of the proteasome exert a strong effect on the response of CBF and COR transcripts to freezing. To get insight into the functional relevance of these differential responses, we followed the induction of CBF4 transcripts in grapevine species that differ in their freezing response. We find that a swift and strong induction of CBF4 under freezing is a hallmark of tolerance, and that the responses to chilling are uncoupled from responses to freezing. We infer from these data a model where chilling and freezing stress trigger different signal chains, culminating in the differential nuclear import of CBF4, leading to adaptive responses that depend on the type of cold stress.

## 2. Results

### 2.1. VvCBF4 Is Imported into the Nucleus in Response to Cold Stress

The CBF4 proteins of *V. vinifera* clearly differ from the other CBFs, as manifested from a phylogenetic tree inferred for known sequences from different varieties of *V. vinifera*, along with several wild species of Vitis (Appendix A). While CBF1-3 from the reference genome *V. vinifera* cv. Pinot Noir distribute separately with a high bootstrap support of >90%, the CBF4 proteins form a common cluster. Only the two North American species *V. riparia* and *V. cinerea* differ slightly, forming a clade with a bootstrap value of just above 50%. A sequence alignment of the four CBF members from *V. vinifera* Pinot Noir (Appendix A) reveals that the AP2/ERF DNA-binding domain as well as the predicted nuclear localisation signals are highly conserved between all members of the CBF family. In contrast, CBF4 differs from the other members by lacking a domain in the C-terminal half of the protein. This region, spanning around 20–25 amino acids in the other CBF members, is very rich in serines, indicative of a potential regulation by kinases. To obtain insight into the function of VvCBF4, we generated a C-terminal fusion with the Green Fluorescent Protein (GFP) and expressed this fusion construct in tobacco BY-2 cells under control of the constitutive CaMV 35S promoter. Upon transient expression following biolistic transformation and inspection by spinning-disc confocal microscopy, we observed a punctate signal in the perinuclear cytoplasm and the transvacuolar strands that emanate from the nucleus in these vacuolated cells (Figure 1A,B). In contrast, the karyoplasm was void of any signal. Since VvCBF4 (as well as the other members of the CBF family) harboured a bona-fide nuclear localisation signature (Appendix A), we would have expected an intranuclear GFP signal. To address this further, we generated a stable transgenic line expressing the VvCBF4 GFP fusion. When these cells were investigated at a normal temperature (25 °C), we observed again a cytoplasmic signal (Figure 1C). When we zoomed into the GFP channel and compared it with the overlay of the image obtained by Differential Interference Contrast (Figure 1D), the GFP signal was clearly located outside of the nuclear envelope, while no signal was detectable in the karyoplasm, confirming the results from the transient expression. We wondered whether the subcellular localisation might be conditional and then conducted an experiment where the cells were subjected to severe cold stress (0 °C for 4 h). This resulted in a generally reduced intensity of the signal (Figure 1E), but now, the close-ups revealed that the protein was found in speckles localised inside of the nuclear envelope (Figure 1F). Thus, in the absence of cold stress, the GFP fusion of VvCBF4 is cytoplasmic, but seems to be imported into the nucleus when the cells experience cold stress.

### 2.2. VvCBF4 Mitigates Mortality Inflicted by Cold Stress

To address the role of VvCBF4 in cold tolerance, we followed cell mortality under continuous cold stress either in the BY-2 cell line expressing VvCBF4 in a stable manner or in the non-transformed wild type (Figure 2). When the wild type was kept at 0 °C, mortality increased steeply after 24 h of induction, increasing 3 to 4-fold over the resting level within 72 h. For the cells expressing VvCBF4, this cold-induced mortality was clearly mitigated by around 40% at 72 h, while the mortality of unchallenged cells was comparably low for both lines. In the next step, we followed mortality under chilling stress (4 °C). Here, a significant increase in mortality was observed much later (after 72 h of stress treatment) and at a lower amplitude (less than 2-fold over the resting level at 72 h of chilling). Again, the cells expressing VvCBF4 seemed to be sturdier, but the difference was less pronounced (by around 20% at 72 h) and did not reach significance. Thus, the expression of VvCBF4 mitigates mortality inflicted by cold stress.

To get insight into underlying mechanisms, we measured extracellular alkalinisation as a proxy for calcium influx (Appendix A). In the non-transformed wild type, the pH increased swiftly upon transfer to ice water; at the first measured time point at 20 min, it had already reached 0.4 units and continued to grow to almost 1 entire pH unit until 90 min. To test whether this response was still physiological, we returned the cells to 25 °C at this time point and observed that the pH dissipated again, indicating that these cells were still actively regulating their ionic balance. The VvCBF overexpressor was much less responsive; here, the maximal shift in pH was only around 0.5 pH units and the peak was earlier than the transfer to the warm condition. Likewise, the alkalinisation dissipated more swiftly and more thoroughly after the cells had returned to 25 °C. Thus, survival in the cold seems to correlate with a restrained initial calcium influx. Oxidative burst in mitochondria has often been associated with stress-related cell death. Therefore, we measured the production of mitochondrial superoxide using the fluorescent dye MitoSOX by means of quantitative image analysis (Appendix A). While we were able to detect oxidative burst at around 12 h after the onset of stress, we did not see any response earlier, no matter whether the cells were challenged by chilling (4 °C) or by freezing (0 °C). Moreover, the increase was of similar amplitude in both cell lines. For very prolonged chilling (72 h), the signal was slightly more pronounced in the overexpressor as compared to the wild type. Thus, oxidative burst in mitochondria was a late event that, in addition, did not show any correlation with mortality.

### 2.3. VvCBF4 and Its Endogenous Homologue Are Regulated Differently in Tobacco BY-2

To understand why the transfected VvCBF4 mitigated cold-inflicted mortality in the tobacco BY-2 recipient, we followed steady-state transcript levels of *VvCBF4* along with transcripts of the tobacco CBF4 paralogue *Avr9/Cf9* (NP_001312746.1) over time in response to both chilling (4 °C) and severe cold stress (0 °C). The resting level of *Avr9/Cf9* was identical in non-transformed BY-2 cells and cells overexpressing VvCBF4 (Figure 3A), indicating that the overexpression left the expression of the endogenous CBF4 paralogue unaltered. The resting level of the transfected *VvCBF4* was around 6.5 times higher than the endogenous *Avr9/Cf9*, which is to be expected since VvCBF4 is driven by the constitutive Cauliflower Mosaic Virus 35S promoter. Interestingly, the steady-state level of *VvCBF4* transcripts increased significantly when the overexpressor cells were exposed to cold stress (Figure 3B). This induction was more pronounced for chilling as compared to severe cold stress. For chilling, the *VvCBF4* transcripts increased by 3.5-fold of the resting level (corresponding to around 25 times the resting level of *Avr9/Cf9*), but they only increased by 2-fold for severe cold stress (around 13 times the resting level of *Avr9/Cf9*).

In the next step, we investigated the regulatory pattern for the tobacco paralogue of CBF4, *Avr9/Cf9*, under the same conditions. Irrespective of the absence (Figure 3C) or the presence (Figure 3D) of the transgene, there was a strong induction of *Avr9/Cf9* transcripts, peaking at around 200-fold of the resting level at 48 h after the onset of severe cold stress (0 °C). For chilling stress, only a relatively minute induction of less than 10-fold over the resting level resulted. Thus, the temperature dependency of the foreign VvCBF4 (stronger induction at 4 °C compared to 0 °C) was inverted to the pattern seen for the endogenous *Avr9/Cf9* (weaker induction at 4 °C compared to 0 °C). The regulation of the endogenous *Avr9/Cf9* was not affected by the overexpression of VvCBF4.

### 2.4. VvCBF4 Modulates the Expression of Specific CBFs and CORs Depending on Temperature

The transcripts of the foreign VvCBF4 were upregulated under cold stress (Figure 3B), which was accompanied by a reduced mortality (Figure 2). To get insight into the potential functions of VvCBF4 during the response to cold stress, we followed the expression of the endogenous CBF factors *DREB1* and *DREB3* and the three COR transcripts *NtERD10a*, *c*, and *d* (Figure 4).

Since the expression levels of the transcription factors and the COR genes were quite different, we used relative steady-state transcript levels, as inferred by the 2^−ΔCt^ method, to allow comparisons between the different genes, rather than using the representation by fold changes (2^−ΔΔCt^) in Figure 4. The differences between *DREB1* and *DREB3* were quite remarkable (Figure 4). While *DREB1* transcripts did not display any noteworthy changes, neither in the non-transformed wild type, nor in the VvCBF4 overexpressor and neither for freezing, nor for chilling, there was a conspicuous response of DREB3 transcripts. These increased strongly 24 h after the onset of freezing stress, while there was no induction in the case of chilling stress. This accumulation of *DREB3* transcripts reached a plateau at 118-fold of the resting level from 48 h in the case of the non-transformed wild type, while in the VvCBF4 overexpressor, it continued to grow further to reach 150-fold after 72 h of freezing stress.

Since cold tolerance in plants largely relies on the induction of COR proteins that are activated by upstream transcription factors, such as the CBFs, we examined the dehydrin transcripts for *NtERD10a* (GenBank AB049335), coding for a late embryogenesis-abundant (LEA) protein, and *NtERD10c* (GenBank AB049337) and *NtERD10d* (GenBank AB049338), representing splice variants of the same gene encoding a ECPP44-like phosphoprotein. While there was no significant response of *NtERD10a* for any of the cell lines or stress conditions, transcripts for *NtERD10c* and did respond clearly. Transcripts for *NtERD10c* were induced after 24 h of chilling in the non-transformed WT, while in the CBF4 overexpressor, no induction was seen. In the case of freezing, the induction in the WT was slower and of lower amplitude (after 72 h, an induction by 6-fold was observed). In the CBF4 overexpressor, this induction was even further delayed, although the induction at 72 h was comparable to that of the WT (around 8-fold of the resting level). The most salient response was observed for the transcripts of *NtERD10d* (the splicing alternative of *NtERD10c*). These were strongly induced after 24 h of cold stress in both cell lines under both freezing and chilling stress. However, the amplitude of this response was inverted in the two cell lines: in the non-transformed WT, an induction of 8.7-fold was reached after 72 h of freezing stress, while for the VvCBF4 overexpressor, this induction was even stronger (10-fold at 72 h of freezing stress). Under chilling stress, the induction was more pronounced in the WT (14-fold at 72 h), while it was now much weaker (only 4-fold at 72 h) in the VvCBF4 overexpressor. This inverted temperature dependency between the lines matches with the inverted pattern seen for the foreign VvCBF4 transcript (Figure 3B) compared to the endogenous CBF4 orthologue *Avr9/Cf9* (Figure 3C). Taken together, the overexpression of VvCBF4 amplifies the accumulation of DREB3 transcripts in response to freezing stress accompanied by elevated accumulation of the ERD10d, a splice variant of a dehydrin. In the WT, this splice variant is preferentially induced by chilling (not by freezing), correlating with the pattern of the endogenous CBF4 orthologue, *Avr9/Cf9*. In other words, the introduced VvCBF4 imposes the temperature sensitivity pattern of its donor, *Vitis vinifera*, on the COR transcript of the recipient, *N. tabacum*.

### 2.5. Nuclear Import Modulates the Response of CBFs and COR to Cold Stress

As we had observed that VvCBF4 is translocated from the cytoplasm into the nucleus in response to cold stress (Figure 1) and since ectopic overexpression of VvCBF4 altered the response of endogenous CBFs and COR genes to cold stress (Figure 4), we questioned whether translocation of VvCBF4 into nucleus is required for this modulation of gene expression. To address this, we tested the effect of GTP-γ-S, which irreversibly blocks the GTPase function of Ran, and has been shown to also block nuclear import in plant cells [24]. In the absence of cold stress, with the exception of a slight inhibition of DREB1 transcripts, none of the tested transcripts displayed any change (Figure 5). However, under cold stress, the expression of the tobacco CBF4 homologue *Avr9/Cf9* was modulated—the induction under freezing stress was reduced to ¼ after pre-treatment with GTP-γ-S, indicating that nuclear import is essential for this induction by freezing stress. The relatively low induction of the *Avr9/Cf9* transcript by chilling stress was not altered. In contrast, transcripts for *NtDEB1* that are downregulated around 5-fold by freezing stress remain unaltered in the presence of GTP-γ-S. Again, chilling caused only comparatively small effects that were reduced further by the inhibitor. The transcripts for the highly responsive *NtDEB3* were reduced from a 30-fold to a 16-fold induction by GTP-γ-S under freezing stress (under chilling stress, this transcript did not change significantly, neither for the absence nor presence of the inhibitor). Thus, *NtDEB3* behaved in parallel to *Avr9/Cf9*. For the COR gene *NtERD10a*, the inhibitor did not cause significant changes in the response to cold stress. However, *NtERD10c* and *NtERD10d* were altered clearly and with the same pattern by GTP-γ-S. Here, the induction by freezing stress was not changed, but the induction by chilling stress was boosted by around 2-fold. Thus, nuclear import supports the induction of the CBF4 orthologue *Avr9/Cf9* and the CBF *NtDEB3*, while it acts negatively upon the induction of the COR transcripts *NtERD10c* and d.

### 2.6. The COR Transcript ERD10d Is Negatively Regulated by a Protein Synthetised De Novo

To understand the mechanism behind the differential effect of chilling and freezing stress upon the induction of *ERD10d* transcripts (that are derived by differential splicing from the same gene as *ERD10c*), we probed for the effect of Actinomycin D (AMD) as an inhibitor of transcription and Cycloheximide (CHX) as an inhibitor of translation. To find out whether the treatment used in this experiment was sufficient, we first tested their effects on the transcripts of the foreign VvCBF4 that are driven by the constitutive CaMV 35S promoter (Figure 6A). Indeed, inhibition of transcription by AMD significantly reduced *VvCBF4* transcripts to a residual level of around 35% of the resting level, irrespective of temperature. In contrast, inhibition of translation by CHX did not modulate the strong induction of *VvCBF4* transcripts (around 3-fold) in response to chilling stress, nor did it alter the weak induction of these transcripts in response to freezing stress. The pattern for *ERD10d* transcripts (Figure 6B) differed fundamentally. While AMD eliminated the induction of these transcripts irrespective of temperature, there was a clear difference with respect to the effect of CHX. In the absence of cold stress, *ERD10d* transcripts were induced 3-fold by CHX. Likewise, the around 2-fold induction of *ERD10d* transcripts in response to chilling was further accentuated by CHX to around 4-fold of the resting level. In contrast, the strong induction of *ERD10d* transcripts in response to freezing remained mostly unchanged after pre-treatment with CHX. Thus, while transcription is necessary for the induction of *ERD10d*, there seems to be a negative regulator (both in the absence of cold stress and under chilling stress) that is synthetised as a protein de novo. However, this negative regulator is not needed for the response of this transcript to freezing stress.

### 2.7. Different CBF4-Dependent Signalling in Chilling and Freezing Stress

To get insight into the signalling underlying the regulation of cold-related transcripts, we determined transcripts for VvCBF4 (in the tobacco BY-2 cell line VvCBF4ox) and *NtERD10d* (in both, VvCBF4ox and non-transformed BY-2 cells) after 24 h of either freezing (0 °C) or chilling (4 °C) stress after pre-treatment with different inhibitors (Figure 7). The two stress treatments alone induced a different response in the two transcripts—VvCBF4 was induced by chilling but not by freezing stress, while *NtERD10d* was more induced by freezing stress and less by chilling stress (Figure 7, mock). While the response of VvCBF4 did not show obvious changes in the presence of the solvent (1% DMSO), that of *NtERD10d* was clearly modulated. The induction by freezing stress was partially silenced by this solvent in the VvCBF4ox line, while it was significantly enhanced in the non-transformed WT. For chilling stress, DMSO clearly enhanced the induction of *NtERD10d* in both cell lines. Thus, DMSO, which is not only a solvent but also a membrane rigidifier, had a clear effect that was dependent on the type of cold stress and the overexpression of VvCBF4. Pre-treatment with Diphenyliodonium (DPI), an inhibitor of the NADPH oxidase respiratory burst oxidase homologue, significantly quenched the induction of *VvCBF4* transcripts by chilling stress, while *NtERD10d* was not affected (neither in the overexpressor line or the non-transformed wild type). Inhibition of MAPK signalling by PD98059 suppressed the chilling response of *VvCBF4* transcripts to an even greater extent. Here, we observed a mild (in the overexpressor line) or strong (in the WT) induction of *NtERD10d* transcripts. Again, the responses to freezing stress were modulated much less. The most salient changes were seen with MG132, a specific inhibitor of the proteasome. Here, both *VvCBF4* and *ERD10d* were strongly induced under chilling stress (*ERD10d* in both overexpressor cells and WT). A comparable induction of *ERD10d* transcripts was seen for freezing stress, while VvCBF4 transcripts did not exhibit this induction. Overall, the induction of *VvCBF4* transcripts by chilling not only requires RboH and MAPK signalling but is also boosted by a factor that is swiftly degraded by the proteasome. This factor does not play a role in freezing stress. This factor also boosts the accumulation of *ERD10d*, as opposed to *VvCBF4*, during freezing stress as well. The solvent DMSO is inducive for *ERD10d*, especially under freezing stress. This induction under freezing stress was silenced by the overexpression of VvCBF4. Instead, the inhibition of MAPK signalling induced ERD10d under chilling but less so under freezing. This induction under chilling stress was silenced by the overexpression of VvCBF4. These data show that the response of the COR transcript depends on the presence of the foreign *VvCBF4* gene, and that the signalling differs qualitatively between chilling and freezing stress. The most striking trait is the strong dependence of the transcripts on the proteasome because its inhibition by MG132 strongly enhanced the expression of both cold-related transcripts under chilling stress, in the case of *ERD10d* also being under freezing stress.

### 2.8. The Lipoxygenase Inhibitor Phenidone Induces VvCBF4 and VvCBF4-Dependent NtERD10d

The strong effect of the proteasome inhibitor MG132 (Figure 7) led us to the question of whether this effect might be related to the continuous turnover of ICE1. The transcriptional activity of this master switch was shown for *Arabidopsis* thaliana to be modulated by physical interaction with specific members of the JASMONATE ZIM-DOMAIN family [25]. These proteins are negative regulators of jasmonate signalling but are produced in response to jasmonate signalling. If this very specific hallmark of cold signalling was conserved in tobacco cells, we should see specific changes in the cold response when the production of jasmonates is disrupted. This can be achieved by phenidone, an inhibitor of lipoxygenase, thus blocking the conversion of α-linolenic acid into the jasmonate precursor 13-HPOT [26,27]. The solvent control, 1% ethanol, exerted some effect as well; *VvCBF4* expression was induced by 2-fold at 25 °C (Figure 8A). Likewise, at 25 °C, *NtERD10d* expression was induced by 4-fold in the VvCBF4 overexpressor and 13-fold in the WT (Figure 8B). Under cold stress, this effect of the solvent was not noted with the exception of a slight but significant enhancement for the freezing response of *NtERD10d* transcripts. However, the effects of the solvent were minor as compared to the effect seen for phenidone, which already stimulated the transcripts of VvCBF4 by 4-fold under room temperature (Figure 8A). The transcripts of *ERD10d* were also elevated to a similar degree (in the WT by around 4-fold, in the VvCBF4ox by around 6-fold) (Figure 8B). Under chilling, phenidone did not yield conspicuous effects; for VvCBF4, transcript levels were even exactly the same as in the mock control (Figure 8A). The situation differed drastically for freezing stress. Here, *VvCBF4* was induced 15-fold (Figure 8A). For *NtERD10d*, the induction was even more pronounced, but only when VvCBF4 was overexpressed. Under these conditions, *NtERD10d* was induced more than 50-fold (Figure 8B), compared to around 10-fold in the absence of phenidone. Interestingly, this induction was not seen in the wild type (lacking the foreign *VvCBF4* gene and, thus, the induction of *VvCBF4* transcripts by phenidone (Figure 8A)). Taken together, we can show that jasmonate modulates the induction of *VvCBF4* and *ERD10d* transcripts under freezing stress. This effect of jasmonate is repressive since the inhibition of jasmonate synthesis by the lipoxygenase inhibitor phenidone strongly amplifies the induction of both transcripts by freezing stress. The fact that there is no such induction of *ERD10d* seen in the wild type indicates that the endogenous CBF4 orthologue *Avr9/Cf9* cannot functionally replace the foreign *CBF4* from grapevine.

### 2.9. Temporal Cold-Induced Expression of CBF4 Correlates in Freezing Resistance

To get insight into a possible link between CBF4 expression and cold tolerance, we followed CBF4 transcripts in leaves from different Vitis genotypes differing in their cold tolerance, either under chilling (4 °C) or severe freezing (−20 °C). Under chilling stress (Figure 9), the cold tolerant *V. amurensis* from Northern China responded by a rapid (within 3 h) and persistent (over 12 h) accumulation of *CBF4* transcripts. In contrast, *V. vinifera* Pinot Noir from Europe did not display any response of *CBF4* to chilling. Interestingly, the cold sensitive *V. coignetiae* from Japan showed a significant elevation of *CBF4* transcript levels at the first time point but did not produce any significant induction during subsequent chilling stress. The patterns for severe freezing stress (−20 °C) were significantly different. Here, *V. amurensis* accumulated *CBF4* transcripts slightly more sluggishly when compared to chilling stress, but, again, more persistently with a maximum at 12 h. In contrast, *V. vinifera* cv. Pinot Noir, which had not responded at all to chilling stress, produced a rapid, strong, but transient induction of *CBF4* transcripts, peaking at 6 h with a complete breakdown afterwards. The cold susceptible *V. coignetia* was not responsive at all in the beginning and accumulated *CBF4* transcripts only after 12 h. Thus, the cold tolerance of *V. amurensis* correlated with a temporal pattern, where *CBF4* transcripts were not induced very swiftly (no response after 1 h of cold stress) but persistently (over 12 h or beyond), irrespective of the type of cold stress (chilling versus severe freezing). The intermediate *V. vinifera* Pinot Noir was not responsive to chilling stress, indicative of a poor cold acclimation, while it was not able to sustain expression under freezing stress. The susceptible *V. coignetia* was already challenged by chilling stress, responding to a precocious induction, but failed to respond appropriately under freezing stress. In addition to this genotype-dependency, the response pattern differed with respect to stress stringency.

## 3. Discussion

Over the last decades, the essential role of C-repeat-binding factor (CBF) proteins as regulators of the plant response to cold stress have been established [14,20,21,28,29]. In addition to the usual three members of the CBF family, some species harbour a fourth member, CBF4, that is involved in cold acclimation, including important fruit crops such as kiwi [28] or garden strawberry [30] (Koehler et al., 2012). The expression of CBF4 also correlates with cold tolerance in grapevine [22] and overexpression of CBF4 renders grapevines tolerant to freezing [23]. To get more insight into the cellular mechanisms underlying the activity of grapevine CBF4 during chilling and freezing, we have generated tobacco cells overexpressing grapevine CBF4 as GFP fusion. Due to somatoclonal variation, these lines represent a population of different transformant genotypes, such that the readout for gene expression from such a suspension represents an average over many different genotypes. Therefore, we followed the cold response of CBF4 in grapevine leaves from genotypes differing in their cold tolerance. We show that the import of CBF4 into the nucleus, modulating CBF4-dependent gene expression, depends on the quality of cold stress (chilling versus freezing). We show further that the expression of CBF4 and its downstream gene ERD10d depend on jasmonate and proteasome activity.

These findings stimulate the following questions that are discussed below: By what mechanism is cold signalling translated into a change in gene expression? Which events are shared between chilling and freezing signals and where do they diverge? What is the function of CBF4 in the responses to chilling versus freezing? At what point do the CBF4 homologues from grapevine and tobacco differ and what might be the evolutionary context behind these differences?

### 3.1. Nuclear Import of CBF4 as a Regulator, but Not as the Only Regulator

Low temperature is primarily a physical signal. However, the adaptive response of the plant cells must be of a chemical nature because protective proteins must be generated de novo to shield membranes against cold-inflicted damage and to buffer metabolism, especially redox homeostasis, against fluctuations deriving from the differential temperature-dependency of different metabolic pathways (for a conceptual review see [4]). These responses must be deployed before freezing actually ensues, because membrane damage by ice crystals would result in the immediate death of the cell. Efficient activation of these responses in the preceding period, where temperature approximates the point of freezing, is a hallmark of freezing tolerance, while sluggish activation of these responses do not allow the plant to prepare for the subsequent freezing episode. A mechanism requiring de novo synthesis of a regulator triggering these adaptive responses would be too slow to provide the necessary speed. A mechanism where this regulator is pre-formed but inactive and is then activated without the need for de novo synthesis is more likely. We propose that the nuclear import of CBF4 meets this criterion. We show that the GFP fusion of grapevine CBF4 is localised in the cytoplasm in the absence of cold stress, but it is seen in the nucleus when the expressing tobacco cells are exposed to 0 °C (Figure 1). When we block nuclear import by GTP-γ-S, an inhibitor of Ran, the GTPase controlling nuclear import [24], we block CBF4-dependent transcript responses, for instance, of the transcription factors *NtDEB1* and *3*, as well as the endogenous CBF4 orthologue *Avr9/Cf9*, and the cold responsive transcripts *NtERD10c* and d (Figure 5). These responses were qualitatively dependent on cold stress—under freezing stress, it is the transcription factors that were altered and under chilling stress, it was the COR transcripts.

To understand this contrasting pattern, it is important to link it with the expression of these transcripts between the non-transformed wild type host and the cells that overexpress VvCBF4 (Figure 4). Here, the accumulation of the COR transcripts *NtERD10c* and d in response to chilling was clearly suppressed when the import of CBF4 was blocked by GTP-γ-S, meaning that VvCBF4 suppresses an accumulation of the COR transcripts under chilling. Interestingly, there is no effect of GTP-γ-S on the response of these transcripts to freezing (Figure 5). Instead, these transcripts are more efficiently induced under freezing if CBF4 is overexpressed (Figure 4).

In contrast, the response of the transcription factors is more sensitive to GTP-γ-S under freezing stress, whereby the tobacco CBF4 homologues *Avr9/Cf9* and DREB3 are inhibited, while *DREB1* is stimulated when nuclear transport is blocked (Figure 5). The amplitude of the *DREB3* response to freezing is enhanced under freezing stress (Figure 4), and, thus, parallels the pattern seen for *ERD10c* and *d*. The endogenous CBF4 (*Avr9/Cf9*) is induced independently of whether VvCBF4 is present or not (Figure 3C,D).

The most straightforward model to explain these results is that CBF4, upon nuclear import in response to chilling, is suppressing an accumulation of the COR transcripts *ERD10c* and *d*. In response to freezing, the import of CBF4 stimulates the accumulation of *DREB3* and *Avr9/Cf9*, while suppressing that of *DREB1*. Thus, the nuclear import of CBF4 has a different effect depending on the temperature of the cold stress. Such a sign reversal cannot be explained by a model where the nuclear import of CBF4 is the only switch. There must be a second factor that acts in concert with CBF4 and is differentially activated or deployed, depending on the severity of the cold stress. However, whether the transcript changes of those genes are caused by the inhibition of the nuclear import of other genes needs to be further investigated, in addition to what extent VvCBF4 is imported into nuclei already under chilling stress. The fact that some transcripts are modulated in the VvCBF4 overexpressors already under chilling stress strongly supports a model where import can also proceed, at least to some extent, under chilling.

Finding a transcription factor in the cytoplasm may seem unexpected at first glance. Sterical hindrance by the GFP tag could be a principal explanation for this. However, we can rule this out for two reasons. First, the GFP tag was placed C-terminally, such that the nuclear localisation sequence and DNA binding were not masked [31]; second, and more importantly, the protein was imported in response to freezing stress, demonstrating that it is functional. It is possible that this temperature-dependent nuclear import is a specific feature delineating CBF4 from other CBFs. In fact, in those cases, where the subcellular localisation of CBFs from type 1–3 was addressed, usually by the ectopic expression of GFP fusions in onion epidermal cells, *N. benthamiana* leaves, or in protoplasts, they were seen in the nucleus (Aloe, [32]; Tea Plant, [33], *Deschampsia antarctica*, [34]; *Phyllostachys edulis*, [35]; Soybean, [36]. The activity of these factors does not seem to be brought about by their import. Instead, it is the import of other activating factors that unlocks CBF activity. For instance, specific cytosolic thioredoxins respond to cold-induced oxidative bursts at the plasma membrane by nuclear import and cleave their sulphur bonds that have sequestered CBF oligomers that are then released and trigger the expression of COR genes [37].

Is the shuttling of a transcription factor itself a specific feature delineating CBF4s from the other CBFs? The different mode of activation might be linked to their specific role in cold acclimation. The fact that even in Vitis itself a CBF1-GFP fusion is exclusively seen in the nucleus [38] would support such a model, where the difference in function is reflected as a difference in activation mechanisms. It should be mentioned, however, that the CBF4 from *Arabidopsis* seems to differ from VvCBF4. Here, a CBF4-YFP fusion was preferentially found in nuclei upon transient expression in *N. benthamiana* leaves and *Arabidopsis hypocotyls* [39], albeit the interpretation of the shown images is not conclusive because there is cytoplasmic background and the resolution of those images does not suffice to detect a perinuclear localisation (which was also not the focus of that work).

Regulation of gene expression by signal-dependent import of transcription factors is a phenomenon that is well-known from animal studies. A classic example is the glucocorticoid receptor that is bound to heat-shock protein 90 and a tetratricopeptide repeat-containing protein, but detaches upon the binding of steroid hormones, enters the nucleus, and acts there as a transcriptional regulator (for review see [40]). Also, for plants, evidence for the signal-dependent import of transcriptional regulators as a common regulatory paradigm has accumulated. Activation of the plant photoreceptor phytochrome will release the bZIP transcription factor CPRF2 from its cytoplasmic tether, shifting it from the cytosol into the nucleus [41]. Also, cold stress can send transcription factors to the nucleus, as found for VaWRKY12, a transcription factor from the Siberian wild grapevine *V. amurensis* that repartitions from the cytoplasm to the nucleus in response to low temperatures [42]. Recently, an unconventional class XIV kinesin motor, Dual Localisation Kinesin, was found to shuttle to the nucleus when the microtubules disassemble in the cold [43]. In the nucleus, this kinesin binds specifically to a motif in the promoter of CBF4 and modulates its expression [44]. Thus, our finding that the nuclear import of CBF4 acts as a regulator for cold-dependent transcription adds another example to this type of regulatory paradigm.

### 3.2. Chilling and Freezing Signalling Differ in the Role of Oxidative Burst and Jasmonate

A part of the complexity and even inconsistency about the role of different events in cold signalling seems to arise because the molecular genetics of the chilling-insensitive model *Arabidopsis thaliana* and the chilling-sensitive model for rice are often listed together (for a critical discussion of this problem see [6]). We have, therefore, used the approach to compare signalling evoked by chilling and freezing stress in the same model, tobacco BY-2 cells. By this approach, we are able to see that the dependence of cold-induced transcripts on different signalling events depends on the type of cold stress (Figure 8). In the following, we want to highlight two of these events:

Using *NtERD10d* as proxy for COR gene expression, we observed that the solvent DMSO, which is also a membrane rigidifier, enhanced the induction of this COR transcript under cold stress. Since DMSO evokes a drop in membrane fluidity and, thus, mimicks the cold, the stimulating effect of DMSO under cold stress is straightforward to understand [10,45]. This stimulation is completely (chilling) or partially (freezing) eliminated by Diphenylene Iodonium (DPI), which, in plants, is a specific inhibitor of the membrane-located NADPH oxidase respiratory burst oxidase homologue (for a discussion of this inhibitor see [46]). Thus, an oxidative burst occurring at the plasma membrane is required for the cold-induced induction of COR transcripts happening in the nucleus. A molecular candidate for this transduction event might be thioredoxins, such as AthTrx-h2, that translocate to the nucleus in response to an oxidative burst, unleashing CBFs there from sulphur bridge bondage [37]. A (non-intuitive) implication of this model would be that the overexpression of VvCBF4 would stimulate the import of CBF4 under freezing stress, amplifying the cold response of ERD10d transcripts (Figure 5), while under chilling, the accumulation of these transcripts should remain sensitive to DPI. This is exactly what we observe: for the VvCBF4 overexpressor, the inhibition by DPI was seen only for chilling stress, while for freezing stress, DMSO was not inducive, and DPI (dissolved in DMSO) generated the same transcript level as the mock control.

A second difference in chilling and freezing signalling was seen for the dependence on phenidone, inhibiting the conversion of α-linolenic acid into 13-HPOT, the first committed metabolite of the jasmonate pathway [26]. Here, we see a very strong upregulation of *ERD10d* and *CBF4* transcripts in the CBF4 overexpressor, while this induction of COR transcripts is not observed in the WT (Figure 8). Moreover, this phenomenon is only seen for freezing, not for chilling stress. Thus, jasmonates are negative regulators of CBF4 expression and CBF4-dependent COR expression under freezing stress (Figure 8). A mild induction of CBF4 and ERD10d transcripts was also seen under room temperature, suggesting that the phenomenon is under the control of a process that also acts in the absence of cold stress. A prime candidate is ICE1, which is continuously synthetised de novo and degraded in the warm condition [14], but accumulates when proteolysis is inhibited by post-translational modifications [15]. JA-Ile induces the expression of the JAZ response regulators, specific members (in *Arabidopsis thaliana*, JAZ1 and JAZ4) repress the binding of ICE1 to the promoters of its target genes (including the CBFs) and, thus, downmodulates cold-dependent gene expression [25]. An implication of such a mechanism would be that phenidone should upregulate the expression of *CBF4* and CBF4-dependent COR expression (*ERD10d*). This is what we observe in Figure 8. A further implication of this model is that the inhibition of proteasome activity should upregulate both transcripts. Also, this implication was tested by us and confirmed using MG132 (Figure 7). This inhibitor is well known to suppress cold signalling due to the ubiquitination and degradation of ICE1 by HOS1, followed by a down-regulation of CBFs [47]. Moreover, the elimination of all responses by Actinomycin shows that de novo protein synthesis is needed, lending further support to this model (Figure 6). A third implication of the model would be that the overexpression of CBF4 should override this mechanism because ICE1 is not required. Again, we did not observe the upregulation of CBF4 under freezing (Figure 7), while *ERD10d* transcripts were induced, indicating that, here, the effect of the other CBFs that are controlled by ICE1 kicks in. The fact that MG132 remains effective under chilling stress (Figure 7) provides a further example that signalling depends on the quality of cold stress.

We arrive at a working model, supported by numerous, non-intuitive implications that could be experimentally tested and confirmed, where chilling and freezing stress are transduced by two concurrent signalling chains whose relative contribution depends on the stringency of stress. One chain works through the nuclear import of CBF4 and is not dependent on oxidative burst at the plasma membrane and the second chain works through a signal deployed by oxidative burst and is independent of CBF4. Under freezing stress, CBF4-dependent signalling dominates and under chilling stress, the concurrent pathway is preponderant. Whether this second pathway employs thioredoxins [37] remains to be elucidated in future studies. Although we can show congruence between our findings in tobacco cells and the regulation in grapevine leaves from different species, transgenic grapes overexpressing CBF4 or knocking it down by CRISPR-Cas should be attempted in the future to understand the functional context in more detail.

### 3.3. Survival under Freezing: It Is the Response of CBF4 to Chilling That Matters

Transcripts for the introduced VvCBF4 were strongly induced by chilling and barely induced by freezing stress (Figure 4). The mitigating effect on mortality was, however, primarily seen for freezing stress (Figure 2) because chilling did not lead to substantial mortality, indicating that tobacco BY-2 cells are fairly tolerant to chilling. In congruence with this notion, the transcripts of the endogenous tobacco *Avr9/Cf9* produced only a small increase in the response to chilling. Thus, the behaviour of CBF4 reflected the level of cold tolerance of the respective progenitor species. Since the expression of VvCBF4 was driven by the constitutive CaMV 35S, the elevated mRNA level of *VvCBF4* in the transgenic line (Figure 3) was to be expected. However, the fact that this transcript level increased even further under chilling must be due to post-transcriptional regulation, such as mRNA stability. In other words, transcript stability must be under the control of cold-induced signalling, which was also reported for other CBFs [48].

To get further insight into the physiological context of CBF4 accumulation, we addressed transcript regulation during chilling and freezing in three grapevine species differing in their cold tolerance (Figure 9). What discerns the highly cold tolerant species *V. amurensis* from its more cold-susceptible relatives? Interestingly, it is not the induction of CBF4 under freezing stress—this is far more pronounced in Pinot Noir. In fact, it is the rapid induction of CBF4 during chilling that can serve as a hallmark for superior cold tolerance. In other words, the tough fellows are those that are the most sensitive to chilling. A previous study arrived at a similar conclusion comparing the responses of winter wheat varieties differing in freezing tolerance [7]. Also in that study, a swift and sensitive response to chilling was correlated with the ability to deploy a high degree of freezing tolerance.

As such, it is not low temperatures that render blurred seasonality such a big problem in agriculture, it is the suddenness of temperature drops that lead to damage. A species adapted to the continental conditions of Mandzhuria (*V. amurensis*) has evolved a very efficient signalling mechanism, leading to a swift induction of CBF4, such that cold acclimation can be deployed rapidly and efficiently as a prerequisite to survive in an environment where drastic increases and drops of temperature are common. In contrast, *V. coignetia*, which comes from a much milder and temperate climate, was not selected for such a swift induction. Likewise, *V. vinifera*, that had survived pleistocenic glaciation in refugia in the Mediterranean and the Black Sea, while able to accumulate CBF4 under freezing, is not very responsive to the chilling that under natural conditions precedes a freezing episode.

These evolutionary considerations shift the upstream signalling driving the induction of CBF4 into the focus of interest. Is it the steady-state levels of signalling compounds, such as RboH, or is it amplifications of the primary sensory events at the plasma membrane that account for the strong chilling response in *V. amurensis*? Is it a second, unknown factor that accompanies nuclear import of CBF4 and modifies the transcriptional response, as suggested by the differences in the response pattern for chilling and freezing in the CBF4 overexpressor? The recent finding that the unconventional class XIV kinesin Dual Localisation Kinesin, in response to cold stress, can shuttle from the membrane to the nucleus and specifically modulate the expression of the tobacco CBF4 homologue *Avr9/Cf9* [44] opens new exciting questions on this second signalling pathway and its interaction with CBF4-dependent cold responses.

## 4. Materials and Methods

### 4.1. Cloning and Stable Transformation of VvCBF4 into Tobacco BY-2

Leaves of grapevine (*Vitis vinifera* L. cv. Pinot Noir, clone PN40024, the genotype used for the grapevine reference genome [49]) were collected after exposure to 4 °C for 6 h. cDNA was synthesized from 1 µg of mRNA template extracted via a commercial RNA isolation kit (Sigma Aldrich, Deisenhofen, Germany). Full-length VvCBF4 (GenBank KF582403.1) was amplified with the modified oligonucleotide primers listed in Appendix A for subsequent integration into the Gateway vector pH7WGR2 (Invitrogen, Paisley, UK). This construct allows for the expression of VvCBF4 driven by the CaMV 35S promoter as fusion with green fluorescent protein (GFP) at the C-terminus, such that nuclear import and DNA binding are not sterically impaired by the tag ^2^. The stable transformation was generated by introducing the construct into suspension-cultured tobacco (*Nicotiana tabacum*) BY-2 cells.

### 4.2. Cell Culture and Cold Treatment

Tobacco (*Nicotiana tabacum* L. cv. Bright Yellow 2, BY-2) cells were cultivated as a suspension in modified liquid Murashige and Skoog (MS) medium and sub-cultured weekly, as described in [50]. To administer freezing (0 °C), the entire Erlenmeyer flask with the cells was placed at day 5 after subcultivation into an ice water mixture on an orbital shaker (KS260, IKA Labortechnik, Germany) at 150 rpm in the dark. To administer a chilling treatment (4 °C), cells were kept in a cold room, again on an orbital shaker. If not stated otherwise, the cells remained exposed to the cold treatment for 24 h.

### 4.3. Monitoring the Cell Mortality under Cold Stress

To investigate the effect of VvCBF4 on cold response, cell mortality over a time course was assessed via Evans blue assay, as described in [51]. Cells under freezing and chilling treatment were collected every 24 h to continuously monitor cell mortality. Data from each treatment at one time point were based on at least 4 independent experiments with more than 1500 cells.

Extracellular alkalinisation was used as a readout of calcium influx (using the co-import of protons as a proxy). The changes in extracellular pH were measured by a pH meter (pH 12, Schott Handylab) with a pH electrode (LoT 403-M8-S7/120, Mettler Toledo). The cell suspension (4 mL) was pre-equilibrated on a shaker at 25 °C in the dark for 1 h before administering cold stress. As an additional readout, the intracellular distribution of calcium was measured by the fluorescent dye chloro-tetracycline. The cells were sampled after their respective treatment and then transferred into a mesh-like device to remove the medium. The filtered cells were fixed in 2.5% glutaraldehyde in a 200 mM sodium phosphate buffer (pH 7.4) for 15 min. The fixative was washed out in three rounds (each 5 min) with staining buffer (50 mM Tris-HCl, pH 7.45) and excess liquid was carefully drained out by filter paper, before staining for 5 min with 100 μM chlorotetracycline. Unbound dye was washed out twice for 2 min, and cells were directly analysed by spinning-disc confocal microscopy. The green fluorescence was recorded by spinning disc confocal microscopy upon excitation with the 488 nm line of an Ar−Kr laser (Zeiss) and collection of the green emission. For evaluation, mean fluorescence intensity was quantified using ImageJ. Concerning the GFP emission of the VvCBF4ox line, the fluorescence intensity was corrected by the mean of the initial fluorescence intensity from the untreated VvCBF4ox line as a readout for calcium.

### 4.4. Inhibition of Transcription and Translation

To test for the role of transcription in the regulation of CBF4, the inhibitor Actinomycin D (AMD, Sigma-Aldrich, Deisenhofen, Germany) was used, which intercalates into the DNA and, thus, prevents transcriptional elongation by polymerase I (for review see [52]). To address the role of translation, we used the protein synthesis inhibitor cycloheximide (CHX, Sigma-Aldrich, Deisenhofen, Germany). CHX inhibits eEF2-mediated translocation in eukaryotic ribosomes [53]. Both VvCBF4 over-expressor and non-transformed BY-2 wild type cells were pre-treated for 2 h with either 100 μg/mL CHX or 50 μg/mL AMD under shaking at 150 rpm in the dark, before transfer to 0 °C and 4 °C for a subsequent 24 h. As a solvent control for AMD, cells were treated with 2.5% ethanol. Cells were sampled directly after the pre-treatment, and cells treated with cold only without inhibitor pre-treatment were included as additional controls.

### 4.5. Pharmacological Treatments

To address the role of the NADPH oxidase respiratory burst oxidase homologue (RboH) in cold signalling, cells were pre-treated with 10 µM of Diphenyleneiodonium (DPI) specifically inhibiting the generation of superoxide [2] (Eggenberger et al., 2017). The MAPK cascade was assessed by pre-treatment with 100 µM of the MAP kinase kinase blocker PD98059 [54]. The cold-induced transcription factor cascade is under the control of the regulator inducer of CBF expression 1 (ICE1), a protein that is rapidly generated and broken down under normal temperatures. In contrast, this protein accumulates in the cold because its proteolysis in the proteasome is blocked. To address this, we used 100 µM of MG132, a specific inhibitor of the proteasome [55]. All three inhibitors were purchased from Sigma-Aldrich, Deisenhofen, Germany, and dissolved from stock solutions in DMSO. They were administered 30 min prior to cold treatment, and the cells were incubated at 27 °C on an orbital shaker (KS260, IKA Labortechnik, Germany) at 150 rpm in the dark. After pre-treatment, cells were transferred to 0 °C or 4 °C, respectively, for 24 h on the shaker in the dark at 150 rpm. For the solvent control, the cells were treated with 1% (*v*/*v*) DMSO, corresponding to the solvent concentration in the inhibitor treatments. Additionally, cells without pharmacological treatment under 0 °C and 4 °C for 24 h were added as reference.

### 4.6. Inhibition of Nuclear Import by GTP-γ-S

To address the role of nuclear import, the non-hydrolysable GTP analogue GTP-γ-S was used. This compound also blocks the GTPase function of Ran in plants [24] and, thus, should block a potential nuclear import of VvCBF4ox. Cells were pre-treated for 30 min with 500 μM GTP-γ-S dissolved in distilled water at 25 °C, prior to cold stress (0 °C or 4 °C) for 24 h on the shaker in the dark at 150 rpm. Control cells were treated in the same manner, with the omission of GTP-γ-S.

### 4.7. Inhibition of Jasmonate Biosynthesis by Phenidone

Since JASMONATE ZIM-DOMAIN (JAZ) proteins, the repressors of jasmonate signalling, were reported to interact with ICE1 and suppress its transcriptional activity [25], we used 1-phenylpyrazolidinone (phenidone) to inhibit jasmonate biosynthesis. Phenidone blocks the conversion of α-linolenic acid into the jasmonate precursor 13-HPOT [26]. Cells were pre-treated with 2 mM phenidone at room temperature for 30 min on the shaker in the dark at 150 rpm and subjected to cold stress (either 0 °C or 4 °C) for the subsequent 24 h. As a solvent control, cells were treated with the same volume of ethanol corresponding to the solvent concentration in the inhibitor treatments. Additionally, cells without pharmacological treatment under 0 °C and 4 °C for 24 h were added as mock controls.

### 4.8. Quantification of CBF4 Transcripts in Grapevine Leaves under Cold Stress

To get insight into the functional context of CBF4 induction, we used a series of grapevine species differing with respect to their tolerance. These included the cold-tolerant *V. amurensis* (KIT-voucher 6540), the cold-sensitive *V. coignetiae* (KIT-voucher 6542), and *Vitis vinifera* L. cv. Pinot Noir (KIT-voucher 7474, corresponding to the clone PN40024 that had been used for the reference genome [49]) from the germplasm collection established in the Botanical Garden of the Karlsruhe Institute of Technology. Plantlets were propagated clonally from wood cuttings and were used for this experiment at the age of 10 weeks. Entire plants were transferred either to a chilling treatment (in a cold room at 4 °C) or subjected to a freezer (−20 °C) for specific time intervals (0, 1, 3, 6, 12, and 24 h). Next, the first fully expanded leaves (plastochrones 5–7) were excised, immediately frozen in liquid nitrogen, and then the samples were stored at −80 °C for further investigation.

### 4.9. Quantitative Real-Time PCR (RT-qPCR) Analysis

Steady-state transcript levels were performed using a CFX96 real-time PCR cycler (BIO-RAD CFX96, München, Germany), as described in [8] and calculated according to [56], normalising against L25 as reference gene using the 2^−∆∆Ct^ method. The primers are listed in Appendix A. Data represent mean and standard errors from three independent experiments, each in technical triplicates.

## Figures and Tables

**Figure 1 ijms-23-11417-f001:**
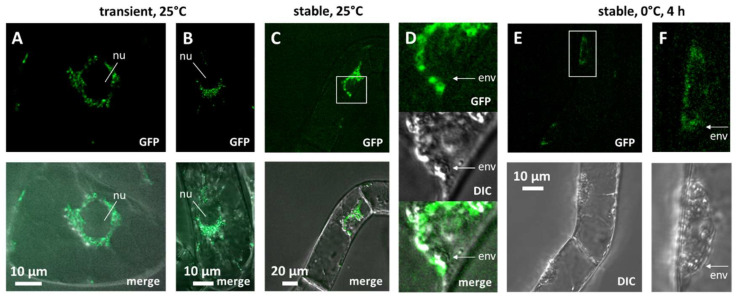
Subcellular localisation of VvCBF4 in fusion with GFP, assessed in tobacco BY-2 cells as a heterologous host. (**A**,**B**) transient transformation with a high-copy vector; cells incubated at 25 °C. For two representative cells, the GFP signals and the merge of the GFP signal with the differential interference contrast (DIC) are shown. Note that the GFP signal is localised outside of the nucleus (nu). (**C**,**D**) stable transformation with a binary vector; cells incubated at 25 °C. (**D**) shows a zoom-in of the region marked by a white square in (**C**). White arrows indicate the position of the nuclear envelope (env) separating karyoplasm and cytoplasm to show that the GFP signal is found outside of the nucleus. (**E**,**F**) stable transformation with a binary vector; cells incubated at 0 °C for 4 h. (**F**) shows a zoom-in of the region marked by a white square in (**E**). White arrows indicate the position of the nuclear envelope (env) separating karyoplasm and cytoplasm to show that, this time, the GFP signal is found inside of the nucleus.

**Figure 2 ijms-23-11417-f002:**
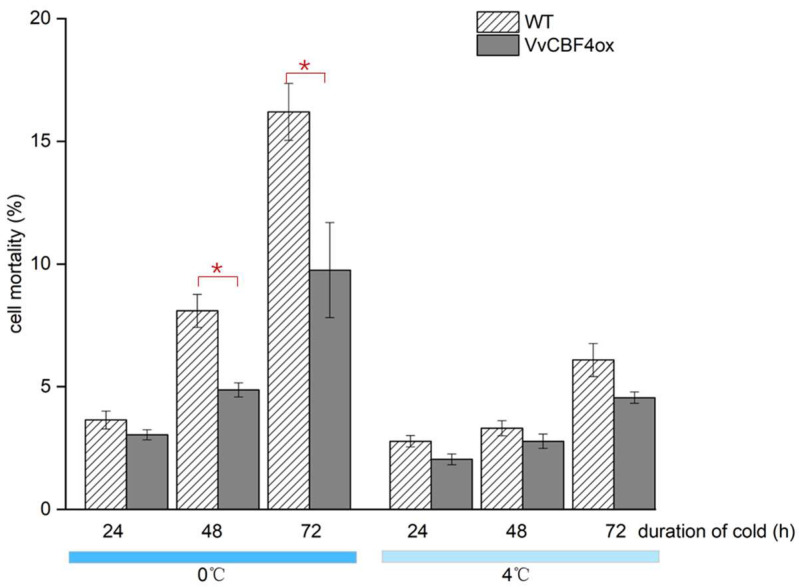
Cell mortality in response to cold treatment (4 °C and 0 °C). Data are shown as means ± SE from three independent experiments with 1500 cells each. Asterisk (*) represents a statistically significant difference with Fisher’s LSD test (*p* < 0.05).

**Figure 3 ijms-23-11417-f003:**
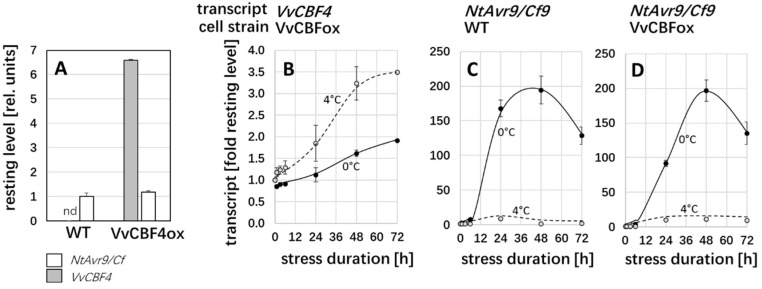
Regulation of transcripts for *VvCBF4* and its tobacco homologue *Nt**Avr9/Cf9* in response to chilling (4 °C) and freezing (0 °C) depending on cell strain. (**A**) The resting levels for the transcripts of the two CBF4 homologues in the absence of cold stress in non-transformed BY-2 cells (WT) versus cells constitutively overexpressing VvCBF4 in fusion with GFP (VvCBF4ox). nd, non-detectable. (**A**–**C**) Time course for steady-state transcript levels for *VvCBF4* (**B**) and native *Nt**Avr9/Cf9* (**C**,**D**) during chilling and freezing in the background of the non-transformed WT (**C**) or cells overexpressing VvCBF4 (**B**,**D**). Data are expressed relative to the resting levels (given in (**A**)) and represent means ± SE from three independent experimental series, each in technical triplicates.

**Figure 4 ijms-23-11417-f004:**
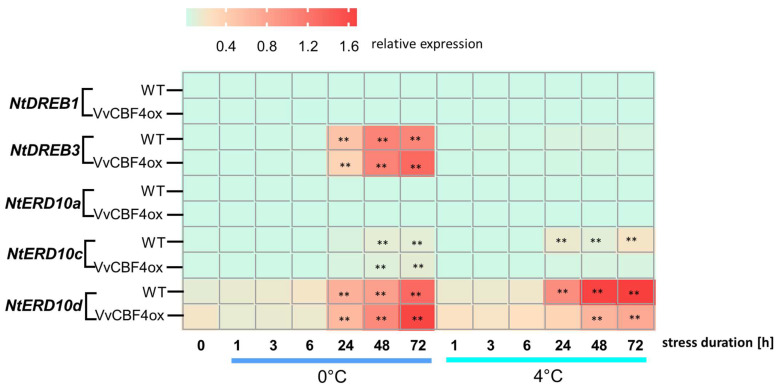
Time course of the relative expression of endogenous CBF genes and COR genes in response to chilling (4 °C) and freezing (0 °C). Colour code shows steady-state levels for two endogenous CBF (*NtDEB1* and *NtDEB3*) and three COR transcripts (*NtERD10a*, *NtERD10c*, and *NtERD10d*) relative to L25, based on 2^−ΔCt^. Data represent means from three independent experimental series, each in technical triplicates. Asterisks (**) represent statistically significant differences in the LSD test (*p* < 0.01).

**Figure 5 ijms-23-11417-f005:**
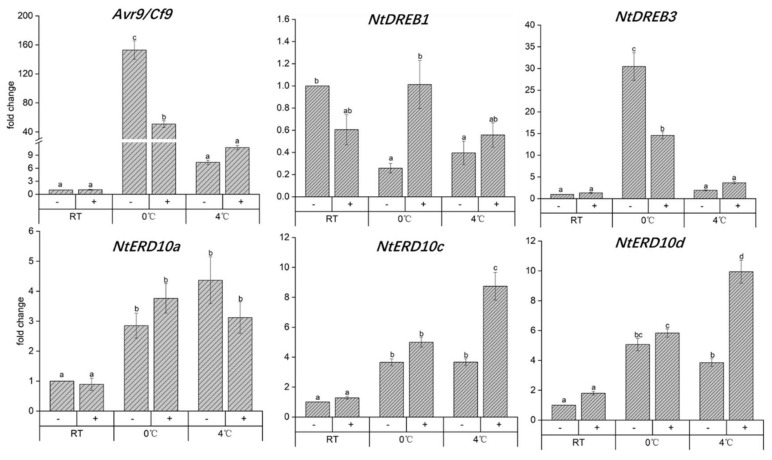
Effect of the nuclear import inhibitor GTP-γ-S on the cold responses of CBF and COR transcripts in the VvCBF4 overexpressor line, measured by quantitative real-time PCR analysis for endogenous CBF and COR genes in response to chilling (4 °C) and freezing (0 °C). Cells were pre-treated with 500 μM GTP-γ-S for 30 min at 25 °C prior to cold stress (0 °C or 4 °C) for an additional 24 h and then examined for gene expression. Controls (-) were treated in the same way, with the omission of GTP-γ-S, while + represents the actual test where cells were treated with GTP-γ-S. Data represent means ± SE from three independent experiments conducted in technical triplicates and are normalised to the untreated control. Different letters indicate the statistically significant difference based on an LSD test (*p* < 0.01).

**Figure 6 ijms-23-11417-f006:**
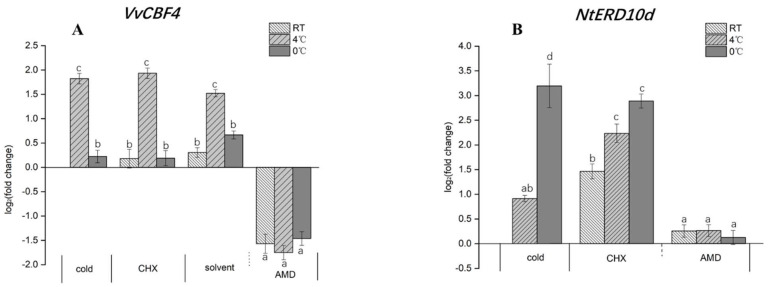
Role of transcription for the expression of *VvCBF4* and *NtERD10d*. Steady-state transcript levels for the foreign *VvCBF4* (**A**) and the endogenous COR transcript *NtERD10d* (**B**) were measured after 24 h incubation of VvCBF4 overexpressor cells and non-transformed BY-2, either at 25 °C (RT), chilling (4 °C), or freezing (0 °C) stress after pre-treatment for 2 h with either the transcription inhibitor Actinomycin D (AMD, 50 µg/mL) or the translation inhibitor Cycloheximide (CHX, 100 µg/mL). The solvent control of AMD was 2.5% EtOH, while CHX was diluted in an aqueous stock. Data are normalised to the expression levels under room temperature without any of the inhibitors and represent means ± SE of three independent experiments in technical triplicate. Statistically significant differences are represented as different letters with Fisher’s LSD test (*p* < 0.01).

**Figure 7 ijms-23-11417-f007:**
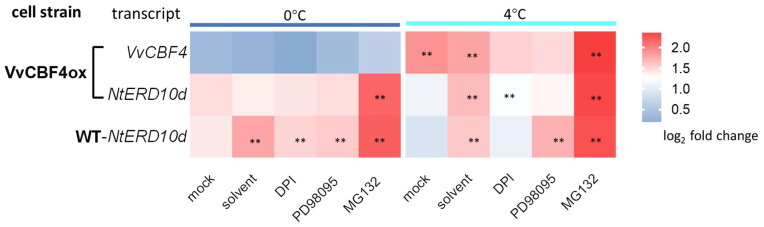
Response of transcripts for *VvCBF4* (in tobacco VvCBF4ox cells) and *NtERD10d* (in both, VvCBF4ox and non-transformed BY-2 cells, WT) to 24 h of freezing (0 °C) or chilling (4 °C) stress after pre-treatment for 30 min with either 10 µM Diphenyleneiodonium (DPI, inhibiting the NADPH oxidase respiratory burst oxidase homologue), 100 µM PD98059 (inhibiting MAPK signalling), or 100 µM MG132 (inhibiting the proteasome). For the solvent control, cells were pre-treated with 1% (*v*/*v*) DMSO in the same volume in treatments. The mock control consisted in an experiment, where the cells were subjected to the respective cold stress for 24 h in the absence of any inhibitor. Data are normalised to the expression levels under room temperature without any of the inhibitors from the three independent experiments in technical triplicate. Asterisks represent statistically significant differences in the LSD test (*p* < 0.01).

**Figure 8 ijms-23-11417-f008:**
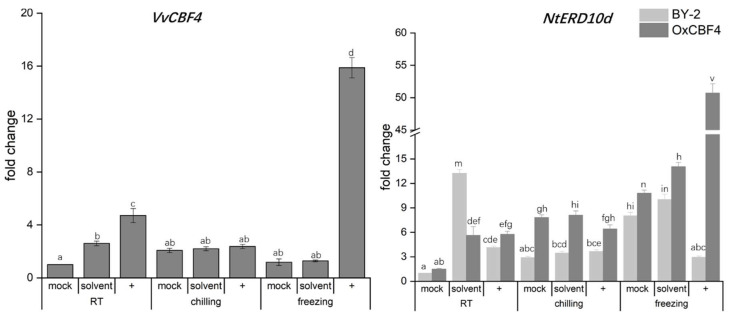
Effect of the lipoxygenase inhibitor phenidone on the expression of *VvCBF4* (in the VvCBF4ox line) and the COR transcript *NtERD10d* (in both non-transformed tobacco BY-2 and the VvCBF4ox line) under 25 °C (room temperature, RT), chilling (4 °C), or freezing (0 °C) stress, represented by fold change (2^−ΔΔCt^ method). Expression was scored 24 h after the onset of the stress treatment either without phenidone (mock) or after pre-treatment for 30 min with 2 mM phenidone (represented by +) or the same volume of ethanol as the solvent control. Data are normalised to the expression levels under room temperature without any of the inhibitors and represent means ± SE of the three independent experiments in technical triplicate. Statistically significant differences are represented as different letters in the LSD test (*p* < 0.01). *NtERD10d* is based on its mRNA level in non-transformed BY-2 cells.

**Figure 9 ijms-23-11417-f009:**
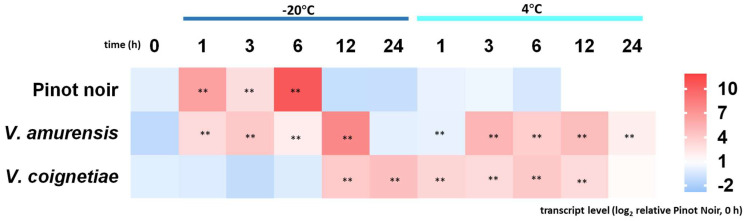
Time course for the response of transcripts for *CBF4* to cold stress in fully expanded leaves from different Vitis genotypes. Plants of comparable developmental states were challenged by chilling (4 °C) and freezing (−20 °C). The leaves were sampled at the indicated time points after cold stress. Steady-state transcript levels are compared to the values found in Pinot Noir prior to cold stress. Data represent mean values from three biological replicates per each genotype and time point. Asterisks (**) represent statistically significant differences in the LSD test (*p* < 0.01).

## Data Availability

All data supporting the findings of this study are available within the paper and within its Appendix A published online.

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
