# Peer review of "Cold-Induced Nuclear Import of CBF4 Regulates Freezing Tolerance"

_ijms, 2022, doi:10.3390/ijms231911417_

Round 1
Reviewer 1 Report
The manuscript by Shi et al., convincingly shows the molecular role of CBF4 in freezing tolerance with physiological, genetic, pharmacological, and gene expression assays. The manuscript is fit for publication in IJMS.
However, I have a few minor comments as stated below:
Title: Better to put it in the form of a Conclusion statement.
Lines 271-287: The supplementary figure has not been cited here. Fig. S2 should be cited properly.
Line 355: The figure shows relative expression but the values do not exceed 1.6 for the maximum value on the scale. How is 8.7-fold related to this scale value? Keep it uniform between text and figure scale.
Reviewer 2 Report
This manuscript reports the function of Grape CBF4 in chilling and freezing using tobacco BY2 cells. The authors found the overexpressing GFP tagged VvCBF4 moved from the cytoplasm to the nucleus by freezing stress and decreased cell mortality under the freezing stress. Besides, they observed that VvCBF4 overexpression impacted the expression of COR genes and investigated a mechanism for gene regulation in VvCBF4 overexpression cells by the pharmacological approach. During the peer review, I noticed the following points, and I hope the points help to improve this manuscript.
Major pints
· The authors analyzed VvCBF4 in Tobacco BY2. It is a heterologous expression system. I understand the difficulty in experiments using a non-model plant, but it seems a bit of a leap to adapt the phenomenon in tobacco culture cells to grapes. In addition, VvCBF4 was overexpressed. The authors should confirm at least one or two data in this manuscript in the grape with a transgenic plant or protoplasts. The authors used grape suspension cells in Wang et al. Plant Science 2019.
· GFP sometimes inhibits protein function. Does the author have sufficient confidence that the constructs used in this study do not inhibit the function of VvCBF1?
· The nuclear import of VvCBF4 is important data in this manuscript. In line 378-379, the authors wrote, “we questioned whether translocation of VvCBF4 into….”. However, the gene expression data in WT is missing in Figure 5. So, the data is insufficient to say that the reduction of Avr9/Cf9 and NtDREB3 at 0 °C with GTP-γ-S is due to inhibition of translocation of VvCBF4, but not any other factor(s).
Minor points
· I could not find the number of independent transgenic cell lines used in this manuscript. How many lines did the authors use to ensure reproducibility?
· Line 271-289: The experiments in this paragraph seem to be the data in Fig. S2. But I could not find it in the main text. So, please add Fig. S2 in the proper place. Besides, please add the explanation of these experiments in the materials and methods if necessary.
· Line 295-317: Correct Fig.4 to Fig.3
· Line 303-304: 35Spro drives VvCBF4 gene. 35S is a constitutive promoter. What mechanism increased VvCBF4?
· Line 309-317: The transcript of NtAvr9/Cf9 at 24 hours shows a significant difference between WT and VvCBFox in Fig.3 D. Why?
· The discussion is a little long and seems redundant. How about writing a little more concisely?
· Figure 1: Why don’t the authors check GFP localization at four ºC? If VvCBF4 controls Avr9 and NtDREB3, it is supposed to remain in the perinuclear cytoplasm.
· Figure 7: MG132 increased the expression of the VvCBF4 and NtERD10d. Is this phenomenon specific to only these genes? Did the authors check other genes?
· Figure 8: Please add A and B to the figure. Furthermore, NtERD10d was expressed more in OxCBF4 than in wild-type BY-2, which is the opposite in the heat map in Fig. 4. Why?
· The homology between VvCBF4 and Ave9/Cf9 seems important in this manuscript. Therefore, the authors recommend adding the Avr9/Cf9 amino acid sequence to the alignment in the Supplemental Figure1.
· The authors used heat maps in this manuscript. While heat map is excellent for capturing the big picture, it does not provide details. Therefore, I suggest the authors provide the standard deviation, standard error, and statistical information of these heat maps in the supplemental data.
Round 2
Reviewer 2 Report
Thank you for your sincere response to my review. I believe the authors responded to the questions well. The manuscript seems to have improved.